# General considerations to experimental research on ocean alkalinity enhancement

Sam Dupont[1,2], Marc Metian[1]

[1] Radioecology Laboratory, International Atomic Energy Agency, Marine Environment Laboratories, Monaco 98000, Monaco
[2] Department for Biological and Environmental Sciences, University of Gothenburg, Fiskebäckskil 45178, Sweden

*Correspondence to*: Sam Dupont (sam.dupont@bioenv.gu.se)

**Abstract.** Ocean alkalinity enhancement (OAE) is proposed as an approach to capture carbon by adding alkaline substances to seawater to enhance the ocean's natural carbon sink. These substances include minerals, such as olivine, or artificial
substances, such as lime or some industrial by-products. Deployment of OAE will lead to complex and dynamic changes in the seawater carbonate chemistry, and in some cases addition of other compounds and impurities from the minerals. While OAE alter the carbonate chemistry in a very different way, much can be learned from the abundant literature on ocean acidification documenting the impact of changes in the carbonate chemistry on marine life from genes to ecosystems. A vast majority of the experimental work was performed by manipulating the concentration of carbon dioxide in seawater under
constant alkalinity (TA) to simulate near-future ocean acidification. Understanding the impact of changes in alkalinity on marine species and ecosystem is less understood. In the context of OAE, it is critical to resolve such impacts, alone or in combination with other compounds and impurities from the minerals to be co-released during implementation, to ensure that any field manipulation does not translate into damaging biological effects. As for other environmental drivers, this will require an understanding across all the levels of biological organizations from species to ecosystems, over relevant time exposure
considering the method of deployment (e.g. dilution, repeated exposure) and factors such as local adaptation. Such complex questions cannot be resolved using a single approach and a combination of monitoring, modeling, laboratory, natural (i.e. proxies or analogs), and field experiments will be required. This chapter summarizes some key general considerations for experimental design. It also compares strengths and weaknesses of the different approaches. We will also consider best-practices relevant to OAE such as the need to properly monitor and consider the addition of trace elements and by-products as
well as the potential interactions with other naturally occurring drivers.

## 1 Identifying a relevant question

A pre-requisite to the selection of a given research approach or strategy is to define a clear question. For a safe and efficient implementation of OAE one needs to answer several key questions, including:

☐  What are the best implementation methods to optimize efficiency and minimize risks?

☐  Is the implementation of OAE safe for marine species and ecosystems?

These questions are too big and complex to be resolved by a single experiment or approach. Fully addressing these would require a large-scale involvement of the scientific community and strong international and multi-disciplinary collaboration. Specifically, in order to fully understand and project the ecological consequences of OAE, a suite of mechanistic based models will need to be developed and connected across all levels of biological organization from species to ecosystem. For example, Dynamic Energy Budget provides a framework to synthesize complex physiological responses and processes at the organism level and allows to project how key traits (e.g. growth, metabolism, reproduction) respond to environmental changes (Kooijman, 2001). At community and ecosystem levels, data are needed on the response of key ecological traits and processes that structure communities, such as predator-prey relationships, competition, habitat provision, facilitation, etc. This will require a wide range of different mechanistic experiments that when combined through parametrization of models will provide the holistic view required for forecasting. These models can then be tested against the response in the 'real' world helping validate the model's underlying parametrization and assumptions.

Exposure to elevated alkalinity at different rates and intensity, potentially combined with the other elements such as silicate, calcium, magnesium and trace metals (e.g. iron, nickel, cobalt, chromium) would expose natural ecosystems to conditions that strongly deviate from the present range of natural variability and has then the potential to drive negative effects. At present, these impacts are poorly understood. Understanding the impact of multiple environmental changes (alkalinity and the consequence for the carbonate chemistry, as well as other elements) on key marine ecosystems requires research at the crossroad between physiology, ecology, and evolution. As a comparison, after more than 2 decades of research on ocean acidification and the publication of more than 10000 scientific articles, we are still lacking the full mechanistic understanding that would allow to bridge chemical and biological changes and the forecasting ability required for science-based management (Cooley et al., 2022).

Regarding the urgency of the climate crisis and the limited resources, it is critical to quickly identify the key sub-questions that need to be urgently answered to provide informed guidance to if, how, where and when OAE should be implemented. These priorities should be identified in the spirit of the United Nation Decade of Ocean Science for Sustainable Development (The science we need for the ocean we want) and focus on the trade-off between the desirable level of understanding to take informed decisions, the time needed to collect such data and how they can lead to concrete actions. Each question can organically translate into a research strategy and the selection of the appropriate approach(es), species/ecosystem, or experimental design (see section 3).

Examples of key sub-questions to resolve the potential impacts of OAE on marine ecosystems include:

- What is the best material (e.g. mineral) for a safe implementation of OAE?
- What is the safest deployment method for the surrounding ecosystems?
- What makes a species or an ecosystem sensitive to OAE?

Resolving these questions would allow to identify the best sites and methods for safe implementation but would require a complex experimental strategy combining laboratory studies (e.g. identify thresholds for key parameters such as alkalinity or trace element concentrations, resolve the combined effect of multiple drivers, develop a mechanistic understanding of how

species and ecosystem resilience (the inherent ability to absorb various disturbances and reorganize while undergoing state changes to maintain critical functions) to OAE links to factors such as present natural variability, taxonomy, physiology, life-history strategies, trophic levels, etc.) as well as field experimentations, including in mesocosms, to validate mechanistic models. That will require to work across the whole range of size and complexity, and break down these complex questions into smaller manageable ones within a strategy.

Additionally, it is important to remember that the implementation of OAE will also involve large-scale industrial activity in marine systems. The impacts of these will be additional to the direct chemical changes and any associated additional stressors with the transport and addition of the alkalinity to the marine system should also be considered.

## 2 Comparison of the different research approaches

Every scientific manipulation experiment, either in the field or in the laboratory, is an abstraction of reality. While best practices, in term of experimental design, measurements or monitoring of environmental conditions, are well-established (see Riebesell et al., 2011 in the context of ocean acidification), the outcome of any scientific study is strongly dependent on experimental choices (e.g. tested scenarios, duration, level of biological organization, selected species or population, etc.) These are often resulting from a compromise between the requested design to test a given hypothesis as well as practical constrains and limitations. Understanding the impact of OAE on marine ecosystem is a complex question that can be broken down into multiple hypotheses. For each hypothesis, a strong scientific strategy involving multiple approaches and/or experiments is needed. In this section, we will briefly describe and highlight the strengths and limitations of each approach (Figure 1).

### 2.1 Laboratory experiments (See Iglesias-Rodriguez et al. (2023), this volume, for more information and references)

Chemical changes associated with OAE deployment can be easily simulated in laboratory experiments. These includes manipulation of alkalinity and/or concentration of the various other compounds or impurities. Different concentrations and dynamic of exposure (e.g. constant vs. fluctuating concentration simulating a dilution; single vs. repeated exposure) can be compared in single or multiple drivers' experimental designs. Laboratory experiments are classically used as a tool to test hypotheses and attribute biological changes to tested variables beyond the correlative approach often used for field observations and manipulations. A wide variety of approaches exist allowing for small to large size experimental units (from mL to $m^3$, depending on the model), single to multiple species or life-history stages, short or long-term exposure, and provide adapted options to work with organisms from bacteria to fish.

Strengths: Experiments in the laboratory offer a wide range of options and have the potential for the highest level of control in the tested parameters (e.g. physico-chemistry, food concentration, species composition, density, etc.) As such, laboratory experiments, in combination with other approaches, are the best alternative to build a mechanistic understanding of the

biological impacts of OAE. While not without limitations, some experimental set-ups allow for a high level of replication and to test complex questions highly relevant in the context of OAE (e.g. what is the biological impact of combined effect of increased alkalinity with trace elements?; what is the biological impact of repeated exposures?). As for any experimentation on living organisms, there are some ethical and sometimes legal aspects associated with biological experimentation. However, those are much easier to resolve than with field approaches.

Limitations: While complex laboratory experiments can have some degree of ecological realism, they cannot fully replicate the complexity of a natural ecosystem. For example, it can be highly challenging to include natural variability for all relevant physico-chemical parameters (seasonal or associated with OAE deployment) or incorporate the full complexity of an ecosystem. As such, mechanistic models developed from laboratory experiments needs to be validated in more realistic settings (e.g. field experiments).

**2.2 Mesocosm experiments** (See Riebesell et al. (2023), this volume, for more information and references)

As for laboratory experiments, manipulations in alkalinity and/or other compounds released during OAE deployments can be performed using mesocosms to achieve a greater level of ecological realism. Mesocosms are generally large-scale enclosed bodies of water, with (benthic) or without (pelagic mesocosms) sediments, including biological communities and related processes that can be experimentally manipulated. Depending on the tested communities, the size can vary between 1L and several $m^3$ of seawater.

Strengths: Mesocosms experiments can partially compensate for the limitations of laboratory-based experiments. They sit between laboratory and field experiments and can be used to evaluate the impact of the tested parameter(s) at the ecological level. Working in a closed system minimizes the public concerns and legal requirement when compared to field trials (GESAMP, 2019).

Limitations: While mesocosms allow for a certain level of controls of the environment, some physico-chemical parameters follow natural variability limiting the ability to attribute the observed effects directly to the tested parameter(s). The size and complexity of mesocosms can also limit the number of replicates and then the ability to detect significant effects. When limitation occurs in term of replication, either in mesocosm or laboratory experiments, an alternative is to replicate by repeating the same experiment multiple times. However, this can introduce unwanted variability as some biological processes vary between days, seasons, and years and decreasing the probability to detect significant effects. Some other limitations include unnatural mixing and turbulence (pelagic mesocosms) or unnatural water flows (benthic mecososms) as well as limitation inherent to a closed system.

**2.3 Field experiments** (See Cyronak et al. (2023), this volume, for more information and references)

Open-system field experiments consist of a direct manipulation (e.g. addition of alkalinity) in a natural system. This approach can be used to simulate an OAE deployment at realistic spatial scale.

Strengths: This approach allows the evaluation of the potential impacts at the ecosystem level in the real world while other environmental parameters naturally fluctuate.

Limitations: Several logistic (e.g. access) and legal challenges (e.g. permit, public acceptance) can be associated with field experiments. Similarly to mesocosm experiments, the cost of the ecological realism is the complexity in attributing the observed effect to the given treatment. It is complicated by the difficulty to truly replicate the experiment and to identify controls. However, this can be partly resolved by substituting space for time and replicating the experiment in time if no strong year-to-year variability is observed.

**2.4 Natural analogs** (See Subhas et al. (2023), this volume, for more information and references)

As for other physico-chemical parameters, alkalinity is not constant across the ocean. The natural variability in alkalinity is linked to cycling of carbon dioxide, calcium carbonate and other minerals. As a consequence, some locations have conditions that can be used as "natural analogs" to OAE deployments. Natural analog sites present environments that resemble the conditions of an OAE implementation and can then be used as a test bed for both sensor deployments and collection of data
on feasibility at scale and potential impacts on key species and ecosystems. These include glacial fjords and runoff into the marine system, seafloor weather of basalts, sites where artificial material are added to the marine system, rivers plumes and deltas, and many others (Subhas et al., 2023).

Strengths: Natural analogs provide the opportunity to work in the field at the ecosystem level and provide a test bed for the interpretation and validation of data collected in laboratory and field experiments as well as models. Different types of analogs
can be used to address different space and time processes (Figure 2 in Subhas et al., 2023) from hours at the deployment site to decades at the global level. Observation in natural analogs also have some practical advantages as it can be less costly than experimental approaches (e.g. mesocosms), logistically risky and does not require complex permits to implement (e.g. field manipulation).

Limitations: OAE analogs have the same constrains as any natural analog for other environmental parameters. While working
in the field provides opportunities for collection of data at higher level of complexity, it lacks the control over the tested variable making it difficult to attribute any observed effect to one or several parameters and does not necessarily account for the presence of impurities or the dynamic of exposure associated with some OAE deployment. While some statistical options are available to disentangle the individual effects of the different environmental parameters (e.g. multivariate and regression analyses), a full attribution is not possible as many non-linear processes and complex interactions are unavoidable when
ecology and multiple stressors are involved. This can be partly solved by incorporating mechanistic understanding and theoretical frameworks coming from more controlled laboratory and field studies. Other limitations include the difficulty of replication and identification of control sites. Natural analogs are also open systems with mobile species flowing through the ecosystem and introducing confounding factors and noise in the collected data.

**2.5 Modeling considerations** (See Fennel et al. (2023), this volume, for more information and references)

The complex scientific questions associated with OAE will require a combination of approaches to develop the needed mechanistic understanding and field validation. Models are critical tools to bridge the different approaches, generate testable hypotheses, upscale from local to global aspects as well as forecast the outcome of different intervention strategies. Developing a fully parametrized model simulating the complexity of the biological response to OAE is extremely challenging. Changes associated with OAE deployment can drive direct effects of each individual driver including impacts of alkalinity, magnesium

and calcium ions on the calcification, or toxic or stimulating effects of trace elements such as iron ions. These can become even more complex and unpredictable when in combination and including the dynamic of exposure. Indirect effects include impacts on the environment properties such as seawater turbidity modulating the propagation of light or cascading ecological processes. A more realistic approach is to use the toolkit of existing models for a fit-for-purpose modelization associated with specific questions. For example, Dynamic Energy Budget (DEB) is one of the most comprehensive frameworks for

bioenergetics, and models based on this theory have been extensively applied to understand the effects of environmental changes, including the ecological consequences (Kooijman, 2001). Ecotoxicological models such as mechanistically based model can be used to resolve the combined effects of the multiple changes associated with OAE deployment (Schäfer & Piggott, 2018).

**3 Best practices: from a scientific question to an experimental strategy**

A full consideration of best practices for experimental design is beyond the scope of this chapter. We will summarize some key general and OAE specific consideration while designing an experimental strategy or experiment. Adapting the famous quote by George Box, we can say that essentially, all experiments are wrong, but some are useful (Field et al. 2015). Each research approach is associated with its own set of strengths and limitations (Figure 1) that combined with practical and technical constrains such as time, space, human resources, money, or expertise, lead to decisions that limit experiments are

wrong, but some are useful the potential of the collected data to answer some complex questions. The full picture can only come from a combination of different approaches and experimental decisions (e.g. Quinn & Keough, 2002).

There are, however, some general best practices that should be followed including the importance of defining proper controls, monitor the physico-chemical parameters following established procedures, including calibration and use of reference materials, use the appropriate level of true replication, and follow best-practices for the measured endpoints (e.g. Riebesell et

al., 2011).

Following best practices optimizes the chance to identify the impact of a given environmental change. Variability is the rule in any biological data and can have different sources: technical (e.g. quality of the method used for the manipulation of a parameter or the measurement of an endpoint), experimental noise (e.g. confounding factors), and biologically relevant (e.g. genetic diversity or driven by the manipulated parameter). Each experiment should be designed to minimize unwanted

variability. This includes randomization of the experimental units, proper training of the person(s) taking care of the experiments, or measuring the endpoints, etc.

For each question and associated experimental design, one must take the following decisions (Figure 2):

- What is my model organism or ecosystem?

One approach is to follow the Krogh's principle: For such a large number of problems there will be some model of choice, or

a few such models, on which it can be most conveniently studied. A given species can be selected for its life-history trait, longevity, physiology, phylogenetic position, sensitivity to the tested parameter, or role in the ecosystem. For example, to study the potential for genetic adaptation to OAE, a species with short generation time would be most appropriate. Model species may be considered when specific techniques are needed (e.g. functional genetics). Additional factors also need to be considered including size, life-history stage, age, weight, sex, etc. Different ecosystems can also be considered, number of

trophic levels, level of complexity, etc.

- Where to sample or perform the experiment?

As a consequence of local adaptation, species and ecosystems evolved different strategies to cope with different locations and environment. For example, different populations of the same species can have contrasting sensitivity to the same changes in

the carbonate chemistry (Vargas et al., 2022). In the context of OAE, physical environment can also influence dilution rates of the alkalinity or the trace elements, the distribution of the particles or the water turbidity, the chemistry can also impact the dissolution of the used minerals, and modulate other drivers or combined effects. The biological characteristics can also influence the potential sensitivity to changes (e.g. natural variability, redundancy, endangered species, other drivers).

- How to design my experimental unit?

To avoid introducing confounding factors, it is critical that the design of the experimental unit (e.g. aquarium, mesocosm) fits the tested species, community, or ecosystem. This includes using the right volume of water, realistic density of biological models, open vs. flow through, density of food, water used, aeration, currents, other physico-chemical parameters, etc.

- How long shall I conduct my experiment or observations?

Based on the question, different durations should be considered to ensure that the observed effect can truly be representative of the treatment. For example, this can be short-term, chronic, or dynamic depending on the tested OAE scenario.

- What is the general experimental design?

Two general experimental approaches can be used: the replicated scenario "ANOVA" approach and the gradient "regression" approach (Figure 3). There are pros and cons to both approaches. The regression approach allows to identify non-linear processes, resolve performance curves, and identify potential thresholds. However, there is the risk of not being able to properly

analyze the collected data if no obvious trend is present. It is also possible to combine both approaches using a collapsed design (Boyd et al., 2018).

- Do I have the proper control(s) and treatment(s) to test my hypothesis?

All research approaches should consider the proper controls taking into account the present natural variability at the relevant spatio-temporal scale as well as conditions in the context of the implementation of OAE. The treatments can mimic a deployment of OAE and cover a wide range of alkalinity (e.g. 1500 to 4000 µmol Kg-1) and other parameters for a more

mechanistic approach. The concentrations of alkalinity and trace elements are not the only parameters to consider as the duration and dynamic of exposure can strongly vary depending on the implementation method. The selection of the experimental approach (laboratory, mesocosm, field, natural analog) and design is highly depending on the question and will directly inform the selection of treatment(s). The OAE dynamics of deployment over space and time is subjected to a variety of physical forcings. The plume dispersal will be influenced by currents, eddies, seabed topography, and other physical

characteristics (Subhas et al., 2023) as well as additional variability from repeated deployments. Any understanding of the biological response to OAE will then need to consider aspects beyond any sensitivity thresholds for alkalinity and trace elements and include the dynamic of exposure. Exposure will vary from immediate "shock" responses at the periphery of a plume to longer-term acclimated responses in ecosystems that may sit directly in the outfall of a plume (Subhas et al., 2023). Some experimental methods may be more adapted to simulate such complex dynamics (e.g. field experiments) as it would

require complex technologies and high level of control and monitoring in closed-systems laboratory or mesocosm experiments. Such complex questions can only be answered through the combination of multiple experimental approaches and a strong communication between fields.

- What to measure?

A wide variety of parameters and methods are available to evaluate biological impacts, including indicators of biodiversity, ecosystem health, and individual fitness. A rule of thumb is to use an endpoint that is as close as possible to the process under evaluation. For example, transcriptomic is often used to infer on organismal physiology while there is very poor correlation between these two endpoints (Feder and Walser, 2005). Ultimately, it is critical to evaluate the potential biological impact of OAE deployment on ecosystem functioning. This will require measuring the impacts at several trophic levels and include the

higher trophic levels. Evaluating the potential ecological impacts is also critical to build trust with local communities. In April 2023, 300 protesters gathered to voice their concerns regarding an OAE deployment in the bay of St Ives and called for greater scientific scrutiny. Specifically, they worried about the impact on the local environment and in particular on the grey seal populations. Seals are benthic feeders that could directly and indirectly be impacted by the heavy metals released (Weeks, 2023).

## 4 Best practices: Specificities to OAE

**4.1 Manipulation of alkalinity** (see Eisaman et al. (2023), this volume, for more information and references)

The desire to increase the alkalinity of aquatic environments is not new and predates the concept of OAE. For example, aquaculture farmers are using liming agents or sodium bicarbonate to restore pond alkalinity to increase photosynthesis and fish production, and to better buffer production water against possible pH changes over time. The so-called "liming" has been used through various materials or chemicals applied in ponds such as agricultural limestone, alkaline slag, agricultural gypsum (calcium sulfate), calcium chloride, slaked lime, quicklime, and lime liquor. While all these compounds mainly neutralize soil acidity before the filling with water, some are more convenient or more effective than others (Boyd and Tucker, 1998). On a smaller scale, aquarists who farm ornamental marine life such as fish, crustaceans, and corals also carefully monitor seawater alkalinity. They use different methods to activate calcium and alkalinity such as additional water changes, kalkwasser (lime water), "balling" and devices such as calcium reactors containing alkaline material that can produce high-alkalinity liquid upstream of the aquarium (Goemans, 2012).

In the context of OAE, different methods of manipulating alkalinity are proposed. Two main options are generally considered:

☐ Addition of ground alkaline material or in situ enhanced weathering.

☐ Pre-dissolution of alkaline materials or agents prior pouring the resulting liquid in studied waters.

These can be directly used in experiments while a more controlled manipulation of the chemistry (alkalinity and other substances) can be used to resolve the mechanisms and modes of action.

When alkaline materials are used, other compounds or impurities can also be released, such as silicate, calcium, magnesium and various trace metals (e.g. iron, nickel, cobalt, chromium). The main elements released through the use of lime, olivine or magnesite are magnesium and calcium ions, along with minor elements like iron and trace elements, that occur at relatively low concentrations in seawater. However, their levels could be sufficient to affect marine organisms (e.g. Hauck et al., 2016; Moore et al., 2013). Therefore, the seawater contamination by the compounds and impurities inherent to alkaline materials has to be properly monitored and included in impact studies.

### 4.2 Monitoring compounds and impurities

There are many analytical methods available for measuring trace metals or other elements. The full process of collecting samples, analyzing dissolved trace elements is time consuming and complex. The existence of multiple chemical forms (speciation), specialized procedures for different elements due to speciation effects and contamination; such analytical work has to be coordinated with specialized laboratories/chemists. One of the major challenges in determining trace metals is indeed preventing contamination of environmental water samples during sampling and analysis (Benoit et al., 1997). Nevertheless, there are some good procedures available online validated by experts to collect and handle samples for dissolved trace elements analysis (e.g. GEOTRACES, 2017; Noble et al., 2020). Among the different research methods discussed in this section, the survey of dissolved trace metals or other elements inherent in alkaline substances in seawater is easier to plan and to realize in

laboratory experiments than in the field as the collection and handling of the samples is more straight forward, and the risk of contaminating samples are much lower.

An exhaustive list of analytical equipment available to analyze all possible compounds and pollutants released into the ocean from each candidate alkaline material is outside the scope of this chapter. The most suitable approach may be to combine a seawater preconcentration system (automated, such as seaFAST or non-automated; Hirata et al., 2000; Wuttig et al., 2019) with inductively coupled plasma-mass spectrometry (ICP-MS). There are exceptions for some elements, but this approach works for most elements expected to be released. Furthermore, the use of passive samplers has the advantage of better temporal and spatial resolution of marine pollution risks compared to discrete samples (Schintu et al., 2014; samples have then subsequently been analyzed in laboratories).

## 4.3 Combined effects of increased alkalinity and compounds and impurities inherent to alkaline materials

Many questions remain to be answered to fully address the potential ecological impacts of OAE and understanding the combined effects of increased alkalinity with other compounds and impurities is a tremendous challenge. Such questions require specific best practices and strategies (Boyd et al., 2018, IOC UNESCO, 2022). Parameters of the carbonate chemistry and other dissolved elements are very likely to have different modes of actions, the functional changes at the cellular and physiological level. Changes in environmental parameters with different modes of action can lead to complex interactions between these parameters, making it difficult to project their combined impacts. Changes in the seawater chemistry can also directly affect the chemical form and bioavailability of a given element (Millero et al., 2009). Resolving these interactions requires a combination of mechanistic studies, modeling and complex multi-stressor experiments.

When considering chemicals such as metals as potential stressors, two different aspects need to be considered. One is the dose-specific effects on the organism, and the other is the complexity of maintaining constant realistic metal exposures in the laboratory.

The relationship between organismal metal exposure and internal dose or adverse effects is nonlinear and depends on the metal studied and the organism selected. The accumulation and storage of bioavailable metals varies widely among aquatic organisms and is element specific. In addition, several metals such as Co, Fe, Mn and Zn are essential for the metabolism of organisms and have optimal concentrations in their tissues (the optimal contents vary from species to species). Therefore, depletion or excess of these elements in an organism can have deleterious effects on the organism (e.g. Forstner and Wittmann, 1983), and some high concentrations may also be beneficial to the organism at certain levels.

From a technical point of view, exposing organisms in microcosms or mesocosms to specific levels of dissolved metals (or mixtures of metals) is more difficult than in field experiments. Indeed, the exposure has to be ideally maintained at a certain level in order to provide a more meaningful risk assessment, but at the same time, it won't fully mirror the reality of the exposure environment due to fluctuations. Furthermore, there is a high likelihood in the microcosm that the presence of organisms with the ability to bioaccumulate metals will decrease exposure levels; repeated doses or flow-through systems will be required to keep the concentration constant.

Nickel may be one of the most important trace metal pollutants in olivine-based ocean alkalization, but there are other potential bioavailable trace metals (such as Cr, Cu or Cd; Bach et al., 2019) which all can be to a certain extent be bioaccumulated (Metian et al., 2007; Hédouin et al., 2010; Eisler, 2009). There is a large body of literature detailing the toxicity, subtoxic concentrations or bioaccumulation potential of many of the compounds release by OAE in marine organisms (e.g. compendium edited by Eisler is one of the most comprehensive sources of information; most elements have an extremely wide range of species - protozoa to vertebrates; Eisler 2009, 2010). However, the effects of some elements found in rocks have not been studied or are poorly reported (e.g., zirconium).

## 5 Key recommendations for experimental research relevant to OAE

Resolving the biological impacts of complex and dynamic changes in carbonate chemistry and other compounds and impurities associated with OAE will require a scientific strategy combining different experimental approaches, methods, and collaboration between disciplines. To successfully develop and implement such scientific strategies, key recommendations include:

- ✓ Identify key scientific questions and sub-questions as well as associated testable hypotheses.
- ✓ For each sub-question, select the most appropriate experimental approach or combination of approaches (laboratory experiments, mesocosms, field experiments, natural analogs, models) as well as locations, biological models, level of biological organization, duration, controls, measured parameters, etc.
- ✓ Follow general experimental best practices for experimental design (e.g. replication, analyses)
- ✓ Take advantage of existing best practices for each specific field involved (e.g. multiple stressors experiments, manipulation and measurements of the carbonate chemistry and/or impurities).

**Acknowledgements**

This is a contribution to the "Guide for Best Practices on Ocean Alkalinity Enhancement Research". The authors, on the behalf of the IAEA, are grateful for the support provided to IAEA Marine Environment Laboratories by the Government of the Principality of Monaco. Authors also would like to acknowledge the invaluable contribution of Philip Boyd, Steve Widdicombe and one anonymous reviewer as their thorough review of the first version of the manuscript helped us to make significant improvements.

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

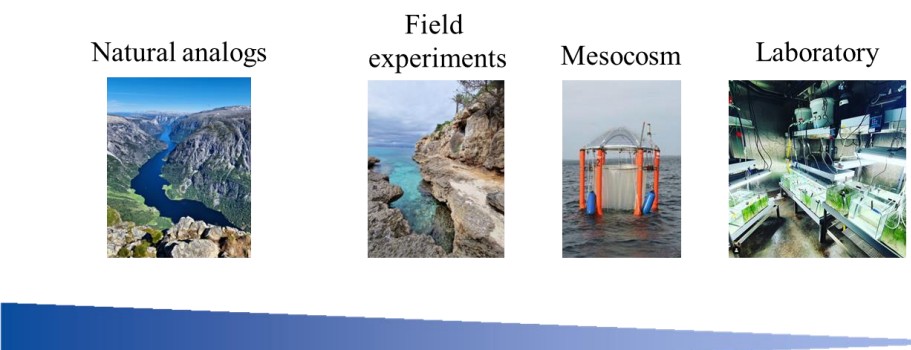

Natural analogs   Field experiments   Mesocosm   Laboratory

Realism
Costs
Practical constrains

Level of control
Mechanistic understanding
Replication

**Figure 1: Simplified version of the strengths and limitations of different complementary research approaches. While the level of environmental and ecological realism decreases from natural analogs to laboratory experiments, field-based approaches are facing other complexities: high costs, legal and practical constrains, lower control and attribution to the tested parameters, lower level of replication. The selection of approach should be based on the question and most questions requires a strategy combining multiple approaches.**

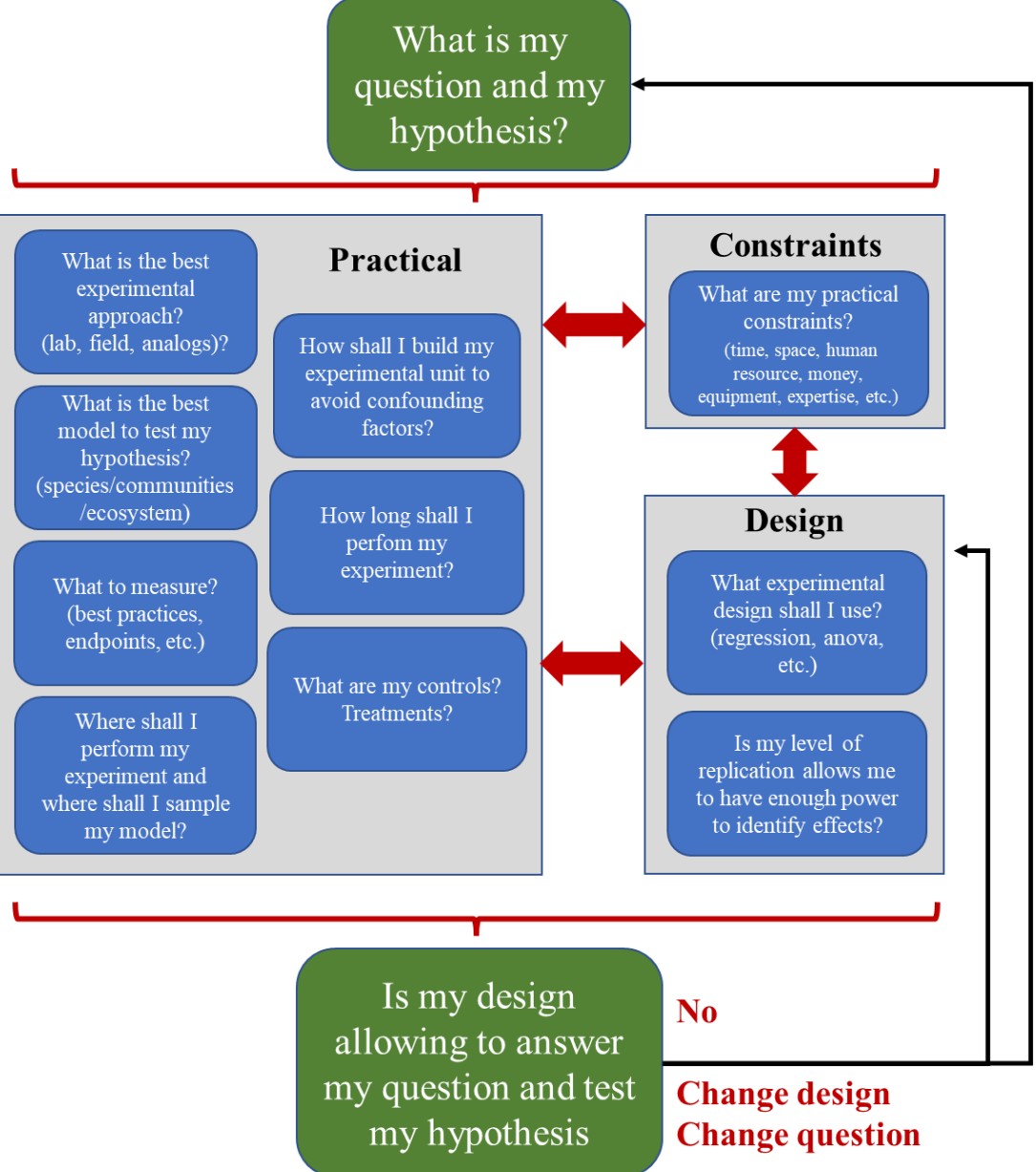

**Figure 2: Flow chart guiding decisions for the design of experiments evaluating the impact of OAE. First, start with a question and a hypothesis. The design of the experiment is an array of decisions at the crossroad between constrains (e.g. time, space, etc.), experimental choice (e.g. tested biological model, duration, etc.) and analytic approach (e.g. regression, ANOVA). When the final design is fixed, ensure that it would allow to answer the initial question. If not, correct your design or, if not possible, change your question.**

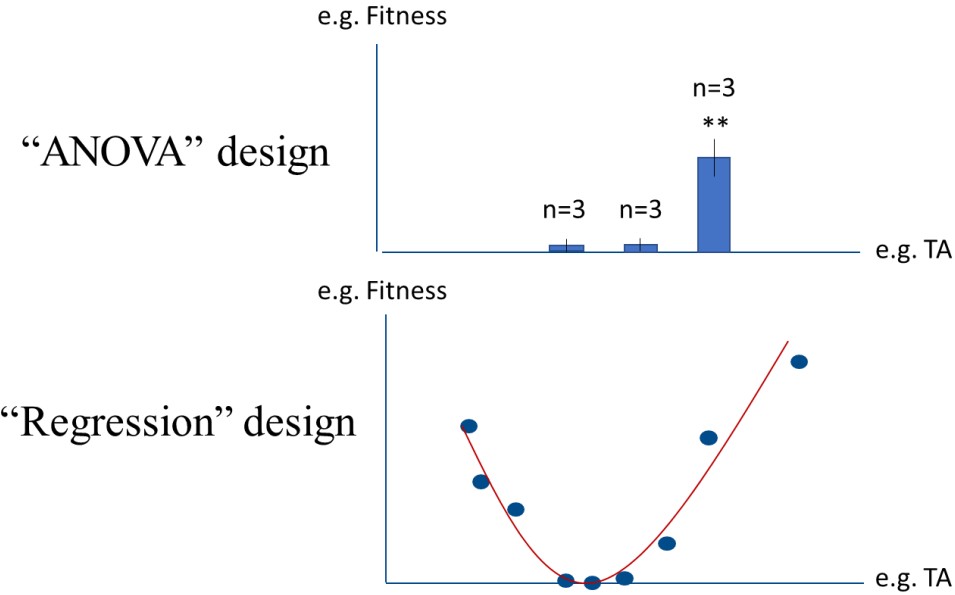

450

**Figure 3: Illustration of two complementary experimental approaches using the same level of replication.**