# Peer review of "General considerations to experimental research on ocean alkalinity enhancement"

_State of the Planet, 2023_

## Author Response (AR1)

Dear Jean-Pierre,

We provided a point by point answer to the reviewers' comments in the online discussion (see copy below) and the corresponding changes are visible in the track change version of the manuscript. We considered all their comments, restructured, corrected and added some sections. As requested, we also changed the title, the way to cite other chapters in the guide and added a recommendation section.

Many thanks for this opportunity and let us know if we missed anything or if you have any question.

Best regards,

Sam & Marc

--

ANONYMOUS REVIEWER

> Although intended to serve as an introduction to Chapter 4, Section 4.1 also stands well on its own, providing a foundation for an understanding of the experimental considerationd of OAE research. From the start, it recognises encourages to reader to think beyond a single experiment, instead highlighting the need for a research strategy. As clearly demonstrated by the authors, without this research *strategy* approach, the "key questions" (section 2) are unlikely to be answered.

**First, we would like to thank the reviewer for this review and encouraging words.**

> The strength of 4.1 is section 2. This section sends a clear message about the need to do these experiments quickly and in a way that informs action. Therefore, I suggest the authors consider swapping the order of section 1 and section 2. As the authors state in lines 95 and 115: identifying the question (current section 2) must happen *before* any decision on the experimental approach (current section 1). In addition, the "key questions" in section 2 (lines 97, 98) provide great context for understanding the pros and cons of the four experimental approaches of section 1, and the last sentence of section 2 would be an appropriate transition to the experimental/modelling approaches of section 1.

**This is an excellent point and we have now swapped the two sections. It makes much more sense as it is also the recommendation that is provided later in the text: always start with a question.**

> Minor suggestions:

- Line 39: Although not directly stated, this sentence implies that a high level of replication is only possible in set-ups with a large number of experimental units. However, a high level of replication can also be achieved by repeating the experiment many times (an often overlooked option). However, I do note that this is acknowledged for a different system later (4.1.3)

**Absolutely, we have strengthened this point by adding this point in the section 1.2 (now 2.2) on mesocosms where the issue of replication is even more of an issue.**

- Sections 1.1 and 1.2: Of the four "approach" sections, 1.3 and 1.4 are the strongest as they start by clearly linking the approach to the issue at hand: OAE. Sections 1.1 and 1.2 would be strengthened by starting with something similar, i.e., 1-2 sentences at start linking the approach directly with manipulation of TA.

**Agreed. We have now added two sentences at the beginning of sections 1.1 and 1.2 (now 2.1 and 2.2).**

- Can section 1.5 be expanded? While the strengths/limitations approach of section 1.2 – 1.4 may not be appropriate here, another paragraph expanding on these concepts would be helpful, especially since so much of the rest of 4.1 focuses on experimental approaches.

**The chapter focus is on experimental approach but we expanded the section to highlight the complexity of predictive modeling and how fit-for-purpose models (e.g. DEB, null models) can be used to resolve simpler questions and/or generate interesting and testable hypotheses.**

- Section 4.1: A link/citation to Chapter 3 would be appropriate.

**Done**

- Line 208: add the charges to Mg and Ca (or spell out)

**Done**

- Finally, there are several typos and grammatical errors, which should be addressed.

**We apologized for these. This manuscript was written in a hurry and we have now revised and corrected the language et formatting.**

**Thanks again for you constructive comments.**

**Sam Dupont & Marc Metian**

--

PHILIP BOYD

> The manuscript is generally well structured and written, but in some places needs attention to better introduce the underlying principles for OAE. The example of what has and has not been achieved in two decades of ocean acidification research (that was international and well-funded) is powerful. See the specific comments below for more details. Specific (to accompany a

marked up pdf sent to the authors to improve the readability of the manuscript and to better explain a range of technical terms)

**Dear Philip,**

**Many thanks for your careful reading, annotated version of the manuscript and your insightful comments.**

> The abstract needs a rewrite. Start with OAE and explain briefly that in most cases it is a mineral (containing various impurities) added to ocean - following crushing - to raise alkalinity as it dissolves. Then introduce analogy with ocean acidification which I think is a robust reality check. Mentioning the impurities – in the mineral phases - up front.  The impurities are important when you get to discussing other elements and trace metals as they introduce complexity with respect to being additional potential environmental stressors. In the case of metals, there influence is dose-specific as they can be beneficial or detrimental to marine life depending on the concentration.

**You are right. Initially, this chapter was not intended to be stand-alone but an introduction to the following sections in the overall guide. We now had modified some part to ensure that it can stand on its own and define all the key concepts. We followed your structure to rewrite the Abstract and strengthen some of these points in the manuscript.**

> With respect to the dose added and also the likelihood of the need for repeated doses, please add more on the importance of dispersion (which is difficult to do in a lab experiment unless you use flow through). Given that at full deployment scales OAE will need to be added repeatedly and over basin scales, how can lab and mesocosm experiments inform the effects of OAE at such scales?  Is there a threshold concentration at which OAE (and its side-effects) must always be below. This warrants more discussion.

**Excellent points. We have now added text in the "Do I have the proper control(s) and treatment(s) to test my hypothesis?" part of the section 3.**

The same goes for bioaccumulation (via repeated additions of OAE).  Co-author is an expert on this topic and at present it is only briefly mentioned.

**Sections 4.2 and 4.3 are now expanded to cover these aspects.**

Also how high in a foodweb must you run texts via experiments. See the St. Ives incident from April 23 and the calls to monitor at apex predator levels. Protesters urge caution over St Ives climate trial amid chemical plans for bay | Cornwall | The Guardian

**This is also an interesting discussion point. We have added some info on that and cited this example.**

**Thanks again, Philip, for your positive and constructive review of this manuscript. It did make it much stronger and interesting.**

--

STEVE WIDDOCOMBE

> Thank you for the opportunity to review this manuscript. The topic of Ocean Alkalinity Enhancement (OAE) is hugely important, and the research community is going to be critical in providing the evidence upon which some enormous decisions for the future, and state, of our ocean will be taken. I look forward to seeing this chapter, and the rest of the special issue, published. Below I have provided some of my thoughts for consideration by the authors which I hope they find useful.

**Many thanks, Steve, for your positive comments on the manuscript and your excellent suggestions. They really helped us to improve this new version.**

> Abstract: Maybe this is more of a stylistic comment than a substantive one, but I would like to see a paper that focusses on OAE not start with a sentence on ocean acidification. I agree that there is much that can be learnt from the past 20 years of OA research, but there is also much that is novel about studying enhanced alkalinity and I don't think OAE studies should be considered as an offshoot of ocean acidification research. Maybe OA research can be seen as an older sibling for OAE, rather than a parent.

**Agreed. All reviewers made similar comments and we have now modified the Abstract, starting with OAE, and adding some more information to ensure that the chapter was "stand alone".**

> Section 1 Comparison of different research methods: This section would benefit from a clear infographic that summarises the attributes of the different research methods and how they complement each other.

**We have the Figure 1 that summarizes the 4 different experimental approaches and their strengths and limitations as a supplement of the text.**

> In my mind I would approach this section the other way around because I would argue that in order to understand and predict the consequences of alkalinity enhancement (or in fact any environmental impact) on individual organisms, populations, communities and, ultimately, whole ecosystems we are going to need a suite of mechanistic based models at each of those levels of biological organisation. Ideally those models will be able to interact with each other to allow us to extrapolate physiological responses all the way up to community and ecosystem responses. For example, at the individual organism level, models that describe an animal's Dynamic Energy Budget provides a framework to synthesise complex physiological responses and processes to predict how key traits (e.g. growth, metabolism, reproduction) will change. This 'whole organism response' will be essential to ensure we capture the inevitable interactions and trade-offs that occur when an animal responds to stress. At the population level we will need to know the impacts on different life history stages of an organism. At community and ecosystem levels, we will need data on the response of key ecological traits and processes that structure communities, such as predator-prey relationships, competition, habitat provision, facilitation etc. So, in order to provide the

information models require, the key question becomes 'what information do we need to identify the mechanisms underlying change and quantify the ways in which those mechanisms respond to OAE?'. The type of experiment can then be designed to specifically supply that required mechanistic understanding. Clearly, no single experiment, or single experimental approach, can provide all the data we need, all have their strengths and weaknesses, so we need to ensure we also have ways to collate, visualise, synthesise and interpret results from many different experiments to gain a holistic view. Whilst experiments are critical for supplying the information required to parameterise models there is also scope for experiments to validate specific outputs of a model 'experiment'. Real world experiments can't operate over the same spatial and temporal scales as those conducted within models, but they can target specific model outputs, simulate the conditions the model used to create that output and test for the response in the 'real' world helping validate the model's underlying parametrisation and assumptions. This interactive relationship between experiments and model simulations will be key to improving our holistic understanding and predictive capability. As a complete aside, in the emerging world of digital twins it would be worth considering the value of using digital twins when undertaking an OAE activity/project in the field. In a situation where the feedback between the rate and scale of alkalinity release and the real-time responses of the ecosystem, and the carbonate system, could be coupled to create a more controlled way to manage an industrial scale release of any substance into the sea. Obviously, this approach will only work if the natural system, and its likely responses, can be reliably monitored and modelled. That is going to need data, and lots of it. So, in summary, I would suggest starting this section by outlining the desired end point. Where do we want to be? Something like, we want to be in a position where we understand the complex and interconnected physiological, behavioural and ecological responses of individuals, populations and ecosystems to enhanced alkalinity and can reliably predict the likely biological consequences of Ocean Alkalinity Enhancement activities. Only then can we have confidence that such approaches are acceptable in terms of their biological impacts. To achieve this we need reliable and effective modelling tools, informed by data and understanding drawn from a variety of experiments and observations.

> Section 2 Identify a relevant question. Based on my points above, it won't be a surprise if I say that I think that the question or hypothesis must be at the heart of any experiment. As the authors point out, the scale and complexity of the question can vary hugely from overarching questions such as "Is OAE safe?" to finely focussed questions that address a small but important part of the story, such as "How does alkalinity enhancement affect copepod respiration rate?". Answering those questions will take very different approaches but they are each important questions to ask in their own right. The key thing is that they are answered robustly, using the correct approaches. I would like to see the section recognise the importance of answering questions across the whole range of size and complexity, not just the 'big' questions. Even their sub-questions are still quite large, complex questions and will require a range of data to answer each of them.

**The Anonymous Reviewer also suggested a re-organization of the manuscript to start with a clearer focus on the ultimate goals before jumping into more technical issues. We have Switched the sections 2 and 1 and it now starts with the section "Identifying a relevant question". We have also incorporated your extra points in a new paragraph, highlighting the importance of mechanistic approach, combination of approaches, field validation, etc. and the need to break down into manageable questions.**

> Section 3 Best practices: from scientific questions to an experimental strategy. I fully agree that this subject is too big for this chapter and could be a whole book in its own right. I wonder whether the best way to summaries this complex topic here would be to present a simple process flow diagram that guides the reader through the various questions they will need to ask themselves as they build their experimental strategy. Such a visualisation could be an exceptionally useful reference diagram to ensure important steps in the process are not forgotten. Maybe there is already something out there in the experimental literature that can be adapted to include the OAE specific issues described in the next section.

**Good idea. We have added a new Figure following your suggestion and based on the text.**

> Section 4 Best practices: Specificities to OAE. Section 4.1 Manipulation of alkalinity. Whilst there are indeed a number of different ways to try and manipulate alkalinity, as you point out, it would be good to get some indication from the current authors which methods they think are most likely to be used in scalable, industrial OAE projects, and therefore should probably be the main focus for future experiments. I appreciate that the authors do not want to limit the choices of experimentalists and therefore present the various options. However, it would be useful to see a more comprehensive description of the pros and cons associated with each method as well as an indication towards what will be the most likely method used in future applications. For example, a recent paper by Hartmann et al (2023) highlighted the importance of adding alkalinity in the right form to actually increase alkalinity but also avoid carbonate formation. They proposed using alkaline solution, particularly those solutions that had been created and equilibrated using reactor techniques to avoid the loss of alkalinity and DIC at the site of application, rather than adding alkaline reactive particles or un-equilibrated solutions. It would seem that this would be a good choice for the majority of future exposure experiments with perhaps other techniques being used when experiments are being used to address specific alkalinity addition processes that propose to use a different application method, e.g. the addition of olivine particles to the surface of sediments. Whatever method an experimentalist chooses, it is essential that the details of the alkalinity enhancement process are adequately reported and the consequences that the chosen process has for the resulting carbonate chemistry parameters are explained and considered when interpreting the biological impact results. [Hartmann, J., Suitner, N., Lim, C., Schneider, J., Marín-Samper, L., Arístegui, J., Renforth, P., Taucher, J., and Riebesell, U.: Stability of alkalinity in ocean alkalinity enhancement (OAE) approaches – consequences for durability of CO2 storage, Biogeosciences, 20, 781–802, https://doi.org/10.5194/bg-20-781-2023, 2023.]

**We agree that it is a key aspect and it is the focus of the chapter Eisaman et al. (2023) in the Guide. This chapter is a general introduction to experimental methods and while we tried to make our chapter stand alone, we don't want to duplicate the focus of other.**

> Section 4.2 Monitoring compounds and impurities. This is a hugely important aspect of OAE. The way in which the alkalinity is delivered could bring along with it significant changes in key ions and other elements that could have as large an impact on organisms as the change in alkalinity itself. For example, we know that changes in the sodium or magnesium concentration can have significant impacts on an organism's physiology. In addition, the impacts on the specification, bioavailability and toxicity of compounds will need to be understood.

**Agreed. As requested by Philip Boyd, we have expanded the section 4.2 and 4.3 to discuss some aspects in more depths.**

> Any Other Business. One thing that I think that is missing from this general introduction is a short paragraph that makes people aware that the industrial scale application of OAE will also involve large-scale industrial activity in marine systems. The impacts of these will be additional to the direct chemical changes caused by the alkalinisation of the water itself. So, for a comprehensive environmental impact assessment of any industrial application of OAE, there will also be a need for sufficient evidence of the likely impacts of the disruptive activities associated with the transport and addition of the alkalinity to the marine system.

**We totally agree with you. However, this is highly technical chapter and we've found it quite complex to add this point as we focus mostly on the direct effects of implementation of OAE. We have added some text mentioning potential additional other effects at the end of the first section but we think that this point would be better developed in other chapters.**

**Many thanks, Steve, for your contribution**

**Sam & Marc**

---

## Author Response (AR2)

Dear Jean-Pierre

We are really please to hear that you agreed with our changes and that the manuscript is accepted. Thanks a lot for your support and work to improve the quality of our manuscript.

We addressed your two last requests:

> Cyronak is misspelt in the body of the text

*Corrected*

> Please add a citation for the quote by George Box

*We have now added a citation for this quote:*

*Field, Edward. (2015). "All Models Are Wrong, but Some Are Useful". Seismological Research Letters. 86. 291-293. 10.1785/02201401213.*

The corrected version is now uploaded

> Your Figure #1 is an illustration using photographic images. Please check if a copyright statement/image credit is required and add it to the figure caption, if applicable. If you are the originator, you can just inform us via email.

*Regarding the pictures used in Figure 1, I confirm that all the pictures are mine.*

Let us know if you need anything else

Best regards

Sam & Marc